# Updating BEM models with 3D rotor CFD data

Marc S. Schneider, Jens Nitzsche, and Holger Hennings

German Aerospace Center (DLR), Institute of Aeroelasticity, Bunsenstraße 10, 37073 Göttingen, Germany *Correspondence to:* Marc S. Schneider (marc.schneider@dlr.de)

**Abstract.** In order to improve the prediction of aerodynamic forces on a wind turbine rotor by the blade element momentum method (BEM), airfoil coefficients are extracted from steady-state 3D RANS simulations of a rotor and then applied for steady-state simulations in a BEM code. The extraction is accomplished by using either averaging of velocities in annular sections, or an inverse BEM approach for determination of the local induction factors in the rotor plane. In this way, 3D rotor polars

- are obtained which are able to capture the rotational augmentation at the inner part of the blade as well as the load reduction by 3D effects close to the blade tip. When using these 3D rotor polars, the radial force distribution from BEM is very close to the RANS result for a variety of load cases, whereas the deviation is often large with 2D airfoil coefficients. However, it is important that the polar extraction is completely consistent the the BEM code in which the polars are supposed to be used. The 3D rotor polars are shown to depend on the blade pitch angle. In addition, the accuracy of the slope of the 3D rotor polars and
- their feasibility for application in unsteady simulations is assessed by a quasi-steady comparison. This work is an updated and expanded version of a contribution to "The Science of Making Torque from Wind" (TORQUE) 2016 in Munich.

# 1 Introduction

Despite the availability of high-fidelity CFD solvers, the Blade Element Momentum method (BEM) is still the state of the art in wind turbine design. In particular in the certification process, where hundreds or thousands of simulations are required, the

- use of CFD simulations would be too expensive with respect to time and computational resources. Instead, it is often tried to improve the BEM by empirical models in order to obtain more realistic results. BEM usually uses two-dimensional airfoil data, which were obtained for cross sections of the blade by wind tunnel experiments or numerical simulations. It assumes that the aerodynamic behaviour of each blade element can be completely described by these 2D airfoil data. It does not account for any three-dimensional spanwise effects. In order to add 3D influence to the BEM, various correction methods for 2D airfoil
- coefficients have been proposed: For a better description of the rotational augmentation of aerodynamic forces and delayed stall close the blade root, models were developed e.g. by Du and Selig (1997), Chaviaropoulos and Hansen (2000) and Dowler and Schmitz (2014). For the decrease of aerodynamic forces close to the blade tip (tip loss), the traditional model by Prandtl/Glauert (described e.g. in Hansen (2008)) is still widely used, although improved and new tip loss model have been proposed (e.g. Shen et al. (2005) or Sørensen et al. (2016)). Bak et al. (2006) present a more general approach which corrects
- 2D airfoil data based on semi-empirical relations for the pressure distribution.

**Figure 1.** The traditional BEM approach with 2D airfoil data (*BEM-2D*) and the BEM with 3D rotor polars from RANS simulations (*BEM-3D*)

Within this work, a set of airfoil coefficients is determined from a number of steady-state three-dimensional RANS simulations of a rotor. This requires the local angle of attack and the lift and drag coefficients to be extracted from RANS solutions. Concepts to do so have already been described e.g. by Johansen and Sørensen (2004) or Guntur and Sørensen (2014). In the present work, two methods for the evaluation of the induction factors and the local angles of attack at the blade were tested and compared. The airfoil coefficients obtained by this procedure will be called *3D rotor polars*. These polars are applied for

and compared. The airfoil coefficients obtained by this procedure will be called 3D rotor polars. These polars are applied for steady-state aerodynamic simulations in a simple BEM code and the results are compared to those of BEM simulations with 2D airfoil data.

Figure 1 shows the traditional concept of BEM with two-dimensional airfoil coefficients as input data (possibly somehow corrected by empirical models; this approach will be called *BEM-2D*), and, in contrast, the BEM with 3D rotor polars extracted

from RANS simulations as input, which will be called *BEM-3D*.

The objective of this work is to treat BEM as a kind of reduced order modelling, in the sense that it is tried to reproduce results from RANS simulations by a drastically simpler and faster model (the BEM-3D). This means that the issue of validation of the RANS solution is not addressed. Instead, the results of the BEM with 3D rotor polars are validated against a number of RANS simulations in the sense of a code-to-code validation.

- A similar work has already been presented at "The Science of Making Torque from Wind" (TORQUE) 2016 in Munich (Schneider et al., 2016). In contrast to the original paper, the present work uses the more renowned reference wind turbine from the European project INNWIND.EU, developed at DTU, with a rotor diameter of about 178 meters and a rated power of 10 MW (cf. Bak et al. (2013)). In addition, section 4 contains some new material in which the difference between steady states is evaluated in order to assess the accuracy of the slope of the 3D rotor polars and their potential for application in unsteady 20 BEM simulations.

#### 2 Extraction of 3D rotor polars from RANS simulations and their use in BEM

#### 2.1 Basic relations

The aim is to obtain airfoil coefficients from steady-state 3D RANS simulations of a rotor. These 3D rotor polars are meant to be used in a BEM code in order to substitute the commonly used polars obtained from non-rotating 2D wings in a wind tunnel or from 2D airfoil simulations. In order to obtain lift and drag coefficients over a range of angles of attack, it is possible to either

- 5 or from 2D airfoil simulations. In order to obtain lift and drag coefficients over a range of angles of attack, it is possible to either fix the rotational frequency of the rotor and vary the wind velocity, or vice versa. Note, however, that these two possibilities are not strictly equivalent, as the centrifugal and Coriolis forces on the blades depend on rotation frequency, but not on wind velocity. One steady-state simulation per wind velocity (or rotational frequency, respectively) produces one angle of attack at each position of the blade and hence one point on the lift and drag polars for each blade element.
- The 3D rotor polars are evaluated from steady-state simulations of an unyawed rotor in a uniform wind field. During the polar extraction, the same assumptions as in the BEM are used: The rotor is assumed to be located in one plane which is perpendicular to the inflow, the rotor blades are discretized into a number of blade elements in the radial direction, and in each blade element, the chord and twist are assumed to be constant. These assumptions are not strictly valid for a realistic 3D geometry, however, the same assumptions will be made later in the BEM in which the airfoil coefficients will be applied.
- First of all, some basic relations from BEM theory are recalled (cf. e.g. Hansen (2008)). From a RANS simulation, the surface forces in the axial and tangential direction  $F_{ax}$  and  $F_t$  on each blade element are known. These forces could be normalized in order to obtain the normal and tangential force coefficients  $C_n$  and  $C_t$ :

$$C_n = \frac{F_{ax}}{\frac{1}{2}\varrho V_{rel}^2 c\Delta r}, \quad C_t = \frac{F_t}{\frac{1}{2}\varrho V_{rel}^2 c\Delta r} \tag{1}$$

where  $V_{rel}$  is the relative inflow velocity at the blade element,  $\rho$  is mass density, c is the chord length and  $\Delta r$  the length of the blade element in radial (spanwise) direction.

However, from a RANS simulation,  $V_{rel}$  is not directly known. Important quantities in the classical BEM theory are the axial and tangential induction factors a and a':

$$a = 1 - \frac{V_{ax}}{V_0}, \ a' = \frac{V_t}{\Omega r} - 1$$
 (2)

where  $V_0$  is the wind velocity,  $\Omega$  the angular frequency of the rotor, r the radial position of the blade element and  $V_{ax}$ ,  $V_t$  are

the axial and tangential velocities in the rotor plane (cf. fig. 2). If the induction factors a and a' were known, the inflow velocity  $V_{rel}$  could be calculated by

$$V_{rel} = \sqrt{V_{ax}^2 + V_t^2} = \sqrt{((1-a)V_0)^2 + ((1+a')\Omega r)^2}$$
(3)

The coefficients  $C_n$  and  $C_t$  refer to the forces normal and tangential to the rotor plane. The lift and drag forces are by definition perpendicular and tangential to the inflow velocity  $V_{rel}$ . In order to obtain the lift and drag coefficients  $C_l$  and  $C_d$ , 30 defined by  $C_l = \frac{L}{0.5 \varrho V_{rel}^2 c \Delta r}$  and  $C_d = \frac{D}{0.5 \varrho V_{rel}^2 c \Delta r}$  (where L and D are the lift and the drag force), a rotation by the inflow

Figure 2. The velocities, forces and angles in a cross section of the blade

angle  $\phi$  is required (cf. fig. 2):

$$C_l = C_n \cos(\phi) + C_t \sin(\phi), \quad C_d = C_n \sin(\phi) - C_t \cos(\phi) \tag{4}$$

where the inflow angle  $\phi$ , in turn, can be calculated from the induction factors by (cf. fig. 2)

$$\phi = \arctan\left(\frac{(1-a)V_0}{(1+a')\Omega r}\right) \tag{5}$$

5

When  $\phi$  is known, the local angle of attack  $\alpha$  can be calculated by

$$\alpha = \phi - \theta_t - \theta_p \tag{6}$$

where  $\theta_t$  is the twist angle and  $\theta_p$  the pitch angle (positive for nose-down; in fig. 2, it is  $\theta = \theta_t + \theta_p$ ).

In summary, if the induction factors a and a' are available, then  $C_l$  and  $C_d$  can be obtained from eqs. (3), (1), (5) and (4), 10 and the corresponding angle of attack  $\alpha$  from eqs. (5) and (6).

The remaining task is now to determine the induction factors a and a' from the RANS simulation. Some methods for that purpose can be found in literature. Shen et al. (2009) obtained the induced velocities and local angles of attack from the pressure distribution or the bound circulation around the blade. Guntur and Sørensen (2014) compared four different methods for the evaluation of the angle of attack from 3D CFD simulations, three of which provide the induction factors as well.

In the present work, two different methods are used to obtain the induction factors: The first one is based on the evaluation of the circumferentially averaged axial and tangential velocity in *annular sections* in a number of slices upstream and downstream of the rotor (method 2 from Guntur and Sørensen (2014); this had already been described before e.g. by Johansen and Sørensen (2004) and is sometimes referred to as *azimuthal averaging technique (AAT)*). The second method is called the *inverse BEM method* (method 1 from Guntur and Sørensen (2014), already described in principle by Lindenburg (2003)).

### 20 2.2 The annular sections method

In this method, the circumferentially averaged axial and tangential velocities are evaluated in annular slices through the computational domain upwind and downwind of the rotor, similar as described by Johansen and Sørensen (2004) (this method is

also referred to as *azimuthal averaging technique*). Then the value of the induction factor in the rotor plane is interpolated for each blade element from the velocities in the last slice upstream and the first slice downstream of the rotor.

From the averaged velocities, the airfoil coefficients  $C_l$  and  $C_d$  and the angle of attack  $\alpha$  are calculated by the following steps:

- 5 1. From the solution of the RANS simulation, calculate the circumferentially averaged axial and tangential velocity component  $V_{ax}$  and  $V_t$  in each annular section
  - 2. Calculate the axial and tangential induction factors a and a' by their definition eq. (2)
  - 3. Interpolate the induction factors to the rotor plane from the last slice upstream and the first slice downstream of the rotor (these slices need to be sufficiently far away from the rotor for not to cut through the walls or the boundary layers of the blades)
  - 4. Use eqs. (5) and (6) to compute the angle of attack  $\alpha$ , and eqs. (3), (1) and (4) to compute the lift and drag coefficients  $C_l$  and  $C_d$

# 2.3 The inverse BEM method

In this approach, the induction factors are obtained from the RANS surface forces iteratively using the same equations as in the BEM. The approach in the present work is very similar to the description by Guntur and Sørensen (2014). The inverse BEM can be described by the following algorithm:

- 1. Initialize a and a', e.g. a = a' = 0
- 2. Compute the local inflow angle  $\phi$  by eq. (5)
- 3. Compute the local inflow velocity  $V_{rel}$  by eq. (3)
- 4. Calculate the normal and tangential force coefficient by eq. (1), where  $F_{ax}$  and  $F_t$  are the sum of all point forces in the RANS solution in the axial or tangential direction, respectively
  - 5. Calculate the induction factors a and a' by the same equation as used in BEM. If the classical BEM algorithm as described by Hansen is used (Hansen, 2008), then it is

$$a = \frac{1}{\frac{4F\sin^2(\phi)}{\sigma C_n} + 1}, \ a' = \frac{1}{\frac{4F\sin(\phi)\cos(\phi)}{\sigma C_t} - 1}$$
(7)

25

10

where  $\sigma = \frac{cB}{2\pi r}$  is the solidity of the rotor with *B* the number of blades, *c* the chord length of the airfoil, *r* the radial position of the blade element, and *F* is the tip loss factor according to Prandtl and Glauert (cf. e.g. Hansen (2008)). If possible, the tip loss correction should be switched of in the target BEM code and *F* should be set to unity; if this is not possible, then at least the same tip loss model should be used in the inverse BEM as in the BEM. This equation is