# Peer review of "Updating BEM models with 3D rotor CFD data"

_Wind Energy Science, 2016_

## Referee Comment (RC1) · Anonymous Referee #1 · 17 Jan 2017

The manuscript is focused on the extraction of three-dimensional airfoil data from blade-resolved RANS analyses of an in-house code. The authors have conducted a great deal of work with respect to the number of CFD RANS analyses, extraction of airfoil tables from RANS results, and comparison of sectional blade forces among results obtained by RANS, inverse BEM, and standard BEM methods. The primary conclusion of the work is that airfoil tables (that account for 3D effects) appear to be dependent on the blade pitch angle when extracted with an inverse BEM methodology that does not include any tip or inboard stall correction. In comparison to a TORQUE paper presented by the authors, the present manuscript also includes an introductory treatment of airfoil data extraction for unsteady cases.

General comments:

The study of three-dimensional effects on turbine blades is an important area of research. In general, the Prandtl/Glauert tip correction (or variants thereof) are used in conjunction with a root loss model and accounting for stall delay effects at the inboard parts of the blade. Contrary to this approach, several researchers have addressed the problem by focusing exclusively on adjusting airfoil data without any tip corrections. Here the reviewer would like to point the authors to literature in the United States on the NREL Phase VI rotor, e.g. the works of Tangler (NREL/CP-500-31243, 2002) and that of Gerber and Tangler published in the ASME Journal of Solar Energy Engineering. Another groundbreaking work in this area is that of Guntur and Sorensen (TORQUE 2012) that the authors are aware of.

While the former method is widely used in the wind energy community for rotor design, the latter method lacks predictive capability due to the fact that airfoil data are extracted from higher-fidelity analyses, e.g. RANS simulations, thus allowing efficient BEM analyses only 'after-the-fact' instead of being an accurate predictive tool for rotor design. The primary purpose of airfoil data extraction methods is to gain further physical insight into a given rotor design and/or to suggest modifications to existing tip corrections towards a more accurate predictive design tool. One has to be cautious here with generalized statements and conclusions because extracted airfoil data are very specific to one particular flow condition on a given turbine blade design.

It is hence not surprising that the extracted airfoil tables may not be applicable to any other rotor designs or even flow conditions (e.g. different blade tip pitch angle) as noted by the authors as their primary conclusion; it is further not surprising that the inverse BEM method agrees very well with the RANS analyses when it is (in reality) a simple verification study that the inverse operations are computed correctly. The authors' main conclusion is that "... 3D rotor polars depend on the blade pitch angle", a result which is (again) really not surprising as the polar extraction method described in section 3.3 does not consider a tip correction which is (in general) a function of the blade flow angle (and intrinsically includes the effect of blade tip pitch) as is well known to the

wind energy community.

As such the results are only of FAIR scientific significance because, in the reviewer's opinion, they only represent a minor contribution to scientific progress in BEM modeling. Furthermore, there are no substantial new concepts, ideas, and methods brought forward by the authors. On the contrary, the description of BEM can be simply referenced by available textbooks (Manwell or Burton) as is the description of the extraction methods documented in Guntur and Sorensen. There are no overall hypotheses, questions asked, or objectives for this work. The reviewer cannot apprehend the actual purpose/objective of this work and how the community can use the results obtained. The scientific quality is also only FAIR because of several statements on rotor induced flow that are erroneous, see specific comments below. The presentation quality is GOOD overall with minor suggestions to rewording.

In summary, the reviewer thinks that this work is of conference proceedings quality but does not merit publication in WES as a selected work from the TORQUE 2016 conference. The reviewer hopes that the authors understand the many technical comments as constructive criticism, and the reviewer would hope to see further work by the authors at future technical meetings.

Specific comments and technical corrections:

Page 1, Line 8: Double "the the . . ."

Page 1, Line 15: A better word choice may be "attempted" (rather than "tried")

Page 1, Line 23: "models" and not "model"

Page 2, Figure 1: Not sure if this figure is needed as it does not provide any new (or more clear) information as presented in the narrative.

Pages 3-4: These are very basic BEM relations that can be simply referenced from Hansen, Burton, or Manwell. No need to go through this.

Page 4, Line 17: Is 2014 the correct publication year?

Pages 5-6: Again, section 2.3 is documented in earlier work [Guntur and Sorensen]. The reviewer does not see the purpose of presenting the analysis/methodology if the reader can simply look at the original publication.

Page 6, Line 12: "... but it is used the other word round" is non-scientific language.

Page 7, Figure 3: It is difficult to discern from the contours but it seems that the axial induction factor a is increasing toward the tip, at least in parts (a) and (c). This is consistent with classical BEM and vortex methods and rejects some of the strong statements made by the authors, see below.

Page 7, section 3.3: Confusing terminology of 'Glauert correction'. The authors refer to it as the high CT correction used in BEM methods, while other literature refers to the classical tip correction. The authors may reconsider to be more clear on this in the future.

Page 7, section 3.3: Reviewer statement: "If no tip correction is used in the inverse BEM analysis, then the 3D polar data MUST depend on the blade pitch angle". In the reviewer's opinion, this is a trivial result and one that is really not useful for BEM-based efficient design tools. The reason why tip corrections exist is because they can account (at least to some extent for lightly loaded rotors) for the effects of solidity (blade number) and blade pitch angle (through the blade flow angle). The reviewer does not see any scientific merit here.

Pages 9-10, section 3.5: The higher the tip pitch angle, the lower the rotor thrust coefficient (which are not documented). It would have been useful to include integrated rotor thrust and power coefficients and to quantify differences beyond qualitative comparisons of sectional blade loads.

Pages 9-10, section 3.5: The authors do not provide a physical reasoning as to the behavior observed in Figure 5 rather than a plain description of what can be seen in the

corresponding figure. The reviewer feels that this is a missed opportunity of providing some further physical insight into the aerodynamic behavior of the given turbine at that particular operating condition. Again, what is the purpose of this work ?

Page 10, Lines 7-9: "It is noted that in the common BEM with 2D airfoil coefficients, the lift and drag polars are always assumed to be independent of the pitch angle. However, the findings of the present work indicate that this is wrong for 2D polars as well as for 3D rotor polars". This is correct but an artifact of the extraction method in the sense that the inverse BEM methodology does not include any tip correction, see also earlier comment. While this is a correct finding by the authors, it is not surprising and, more importantly, does not add much scientific merit to the community as it is very unrealistic to use pitch corrections for airfoil tables in a general predictive sense, particularly when the tip loading is such that the flow is attached (pitch angles of 0deg, 10deg, 20deg). Again, it is not clear what is the actual purpose of this work and what the results should be used for ?

Page 12, Discussion of Figure 7: It would have been really useful to document integrated rotor thrust and torque coefficients to better understand the differences between the various methods. Again, no physical explanation is given; the authors merely discuss what the reader can see in the figures anyhow. Are there any convergence issues for the inboard blade sections ?

Comment on Figures: Vertical axes would be best presented in [kN]. The behavior at the blade root is very concerning; the reviewer suspects that BEM solutions might not be fully converged.

Page 13: Why is load case E2 presented prior to E1 ?

Page 14, Figure 10: The behavior at the blade root is very concerning with respect to numerical accuracy and solution convergence of computed results in general.

Page 14: "... reduction at the tip by increasing the induction factor, which is physically

wrong." and "..., which is physically wrong and can be considered an inconsistency in the tip loss model by Prandtl/Glauert." Unfortunately, these strong statements are both very wrong and concerning as to the authors' understanding of basic blade aerodynamics. 1) The increase in the axial induction factor a toward the blade tip is very consistent with vortex methods (from which the Prandtl/Glauert tip correction is derived) and finite-wing lifting-line theory in general; the reviewer also wants to mention section 3.4.3 in Burton (Relationship between blade circulation and induced velocity) which can serve as a simple counter example to the authors' statements. 2) The reviewer does not understand why the 'a' distribution(s) from Figs. 3 and 6 cannot be used for comparison; in fact, they actually indicate an increase in 'a' toward the blade tip (maybe with the exception of Fig. 3b). This really raises questions as to the scientific quality of this work.

Page 18, Figure 14: What is going on at the blade root ? Are the solutions converged and to what accuracy ? This is very concerning.

---

## Referee Comment (RC2) · Anonymous Referee #2 · 11 Mar 2017

General comments

This paper is very relevant because it discusses the validity of the widely used BEM model and its input. Polars have been extracted from 3D CFD computations and compared to 2D polars from e.g. 2D CFD computations.

However, some important information is missing. We need to know whether the solver includes free transition and how big the CFD domain is. Furthermore, it is assumed that the 3D CFD computations are correct and represents the "truth". Couldn't this be questioned? My experience is that (3D) CFD does not capture stall correctly. So do we believe in the stations where we have separated flow? And forces can be too high if the domain is too small and the flow has been pushed through the rotor because of the boundaries of the domain. The domain boundaries probably have to be 10 to 20 rotor diameters away. Furthermore, the extraction of the data can also be questioned. Was the radial component extracted? Also, there is something with the values where force

distributions are shown. They are about 10 times too high if the forces are Newton per meter.

Finally, the conclusions are not really conclusions but self-fulfilling statements since it is obvious that data based on 3D CFD compares better to 3D CFD.

Since the overall aim of our research is to provide data and methods for the further development of wind turbines, we have to wear the perspective of the wind turbine manufacturers. Would I as a manufacturer blindly believe the 3D CFD computations? I think not. Therefore, I do not think that the advice should be to extract polars directly from 3D CFD. However, I think it could be more interesting to discuss why we see these differences between 2D and 3D. Why do we see this AOA shift for different pitch angles? I am left with the feeling that you (and we) are overlooking something. How do we interpret the abstraction of AOA in 3D flow?

I propose the following should be considered in the paper: * look into the values of the forces, where I think there is an error * describe the CFD setup in more details (domain size, free transition etc) and * analyse where we see differences to 2D data and why you observe the shifts in angle-of-attack * change the conclusions so that it is not self-fulfilling statements. I do not think that you should propose to use polars extracted directly from 3D CFD, because you then make the assumption that 3D CFD is correct.

Specific comments

Abstract Line 8: "...the the..."

Chapter "Introduction" * I think you are missing one of the first attempts to make such airfoil characteristics from CFD: – Bak, C., Fuglsang, P., Sørensen, N. N., Aagaard Madsen, H., Shen, W. Z., & Sørensen, J. N. (1999). Airfoil characteristics for wind turbines. (Denmark. Forskningscenter Risoe. Risoe-R; No. 1065(EN)).

Section "Inverse BEM" Last sentence: I do not really understand...

Section "RANS setup" How big is the domain? How far upstream? Downstream? And in radial direction? If the domain is small this can influence the result. . .

Section "Influence of pitch angle" Figure 5: Title of plot is not clear.

Section "Comparison between Rans,. . ." Fig 7: What does the forces represent? N/m? If so: An integration of the tangential forces result in a power delivered by the blade of around 30MW – and for 3 blade 90MW. . .. Can this be right? A factor of 10? And the same is the case for the axial loading. . .

You extract axial and tangential induction. What about radial? Couldn't this explain some of the AOA shifts?? Or what explains the AOA shift for different pitch angles?

Section "Conclusion" * First finding: – It is obvious that polar data obtained from 3D CFD agrees better to 3D CFD than data not obtained from 3D CFD. So that is not really a conclusion. It have to be so – otherwise you have been inconsistent.

* Second finding: – This is also obvious because you use the inverse BEM. So this is not either a conclusion

* Third finding: – I would actually like to know WHY the polars change with pitch angle. Please consider a bit more.

---

## Author Comment (AC1) · 31 Mar 2017

**Reply to the comment of anonymous referee number 1**

First of all, the authors want to thank the referee for his very detailed comment.

The authors feel that there is a major misunderstanding, probably caused by missing information concerning the purpose of this work in the introduction to the paper, which unfortunately resulted in several misinterpretations by the reviewer. A proposal for a new introduction is appended to this letter of response.

The proposed method is not meant to be used for early design stages (e.g. blade design), but for a later design stage, where the geometry of the rotor is already fixed and an advanced aerodynamic model is desired, e.g. for load simulations during the certification process, where a large number of different load cases have to be simulated with a fixed preliminary rotor design. In this design stage, the effort of performing several CFD simulations for a specific geometry is feasible if the use of CFD-based aerodynamic forces in loads analyses for a specific rotor is desired. The authors should have stated this more clearly in the original version.

Furthermore, it is noted that the purpose of this work is the investigation of a concept for *model updating*, or *reduced-order modelling*, i.e. a fast engineering model (BEM) is updated with data from an expensive high-fidelity method (RANS). The data transferred from RANS to BEM are the 3D rotor polars. This procedure is successful if the BEM with 3D rotor polars produces similar results as the RANS simulations within a certain parameter range. The corresponding paragraph in the introduction will be extended in order to clarify this.

Besides from the new introduction, the authors will revise some other parts of the paper which could lead to misinterpretation. Furthermore, the authors will account for all the valuable technical and linguistic hints from the reviewer.

In the following, the authors have split the reviewer's comment in several items and responded to every single item. The original reviewer's comment is reproduced in *blue italic* letters. Numbering in square brackets has been added by the authors

*[1] The study of three-dimensional effects on turbine blades is an important area of research. In general, the Prandtl/Glauert tip correction (or variants thereof) are used in conjunction with a root loss model and accounting for stall delay effects at the inboard parts of the blade. Contrary to this approach, several researchers have addressed the problem by focusing exclusively on adjusting airfoil data without any tip corrections. Here the reviewer would like to point the authors to literature in the United States on the NREL Phase VI rotor, e.g. the works of Tangler (NREL/CP-500-31243, 2002) and that of Gerber and Tangler published in the ASME Journal of Solar Energy Engineering. Another groundbreaking work in this area is that of Guntur and Sorensen (TORQUE 2012) that the authors are aware of.*
*While the former method is widely used in the wind energy community for rotor design, the latter method lacks predictive capability due to the fact that airfoil data are extracted from higher-fidelity analyses, e.g. RANS simulations, thus allowing efficient BEM analyses only 'after-the-fact' instead of being an accurate predictive tool for rotor design. The primary purpose of airfoil data extraction methods is to gain further physical insight into a given rotor design and/or to suggest modifications to existing tip corrections towards a more accurate predictive design tool. One has to be cautious here with generalized statements and conclusions because extracted airfoil data are very specific to one particular flow condition on a given turbine blade design.*

A reference to Tangler et al. will be added as an example for other approaches based on airfoil coefficients without submodels for tip loss etc.

The 3D rotor polars are definitely specific to the geometry used in the RANS simulations. Therefore, the method is not suitable for blade design. This has been conscious to the authors, although it has unfortunately not been properly documented in the introduction to the original paper.

The method is intended for use in later design stages where the geometry of the blade is already fixed, for instance for load simulations during the certification process for a rotor.

The authors see this work as a contribution in the sense of "model updating" or "reduced order modelling" (as mentioned in the introduction of the paper): A high-fidelity method (RANS) is used to update a fast engineering model (BEM) by means of the 3D rotor polars, which contain the relevant information from RANS. The updated model is then able to produce a result which is very similar to the high-fidelity RANS result. Validation of the RANS result is not necessarily required in order to show the feasibility and limitations of this concept.

The new introduction (see appendix to this letter of response) is hopefully clearer in this respect.

*[2] It is hence not surprising that the extracted airfoil tables may not be applicable to any other rotor designs or even flow conditions (e.g. different blade tip pitch angle) as noted by the authors as their primary conclusion;*

The authors did not perform any investigation regarding the applicability of the extracted airfoil tables to other rotor designs than the one used in the RANS simulations, because in the context of model updating / reduced-order modelling, a good predictive capability outside of the model's scope of application (in this case, for different geometries) can never be expected. The method is intended to be used only for one specific design. Therefore, the investigated methods are definitely not suitable for blade design. It was not the intention of the authors to imply this. This will be stated more clearly in the new introduction (see appendix to this letter of response). The authors, however, think that the dependence of the polars on the blade pitch angle is a finding which is actually worth mentioning, as this is an important fact to consider when using this kind of model updating. The conclusion is not only that there is a pitch-angle dependency, but also that extraction of the airfoil coefficients at a small number of pitch angles (e.g. in steps of 10°) and linear interpolation in between already leads to satisfactory results in BEM-3D. The authors have never seen this statement anywhere in the literature so far, but are grateful for any hints.

*[3] it is further not surprising that the inverse BEM method agrees very well with the RANS analyses when it is (in reality) a simple verification study that the inverse operations are computed correctly.*

This is true for the load cases E1 to E3 presented in the paper, which had been used for the extraction of the polars. In these cases, the excellent agreement of the BEM-3D results with polars from inverse BEM is nothing more than a verification of the method. The authors had already mentioned this in the original paper: "[…] a very good agreement should be expected, as this simply means that the 3D rotor polars, when applied in BEM, reproduce the force distributions from which they were obtained by polar extraction (this can be considered a kind of **verification**)." (p. 11, l. 17ff).

This is not true for the load cases P1 to P3, which actually are prediction for load cases which have not been used for the extraction of the 3D rotor polars. The comparison with the (additional) RANS results in the corresponding figures is a code-to-code validation. Of course, the load cases P1 to P3 are within the parameter range of the extraction cases; this is a general practice in the context of reduced-order modelling as extrapolation to operating points outside of the parameter range of the model identification cannot be expected to yield feasible results.

*[4] The authors' main conclusion is that "… 3D rotor polars depend on the blade pitch angle", a result which is (again) really not surprising as the polar extraction method described in section 3.3 does not consider a tip correction which is (in general) a function of the blade flow angle (and intrinsically includes the effect of blade tip pitch) as is well known to the wind energy community.*

The fact that pitch-angle dependent polars are required for a good match with RANS results is, according to the authors' knowledge, not very well known to the wind energy community. In fact, the authors are not conscious of any publication in which a similar analysis has led to this conclusion (any hint is appreciated). Therefore, the authors think that this is worth mentioning. Regarding the state-of-the-art tip or hub correction models, the authors want to note that significant differences between RANS and BEM are present in all parts of the blade, not only at the root and the tip. Therefore, the authors do not think that a tip or hub loss model can describe the pitch angle dependency accurately, in the sense that it can bring the BEM results close to the CFD results (this, however, is the goal when doing a model update). In addition, please note that the authors have used the tip loss model by Prandtl/Glauert for the BEM-2D analyses, which brought the results only very slightly closer to the CFD results (cf. the figures in Appendix 1 at the end of this letter of response, where BEM results with and without Prandtl's tip loss model are compared).

*[5] As such the results are only of FAIR scientific significance because, in the reviewer's opinion, they only represent a minor contribution to scientific progress in BEM modeling.*

The authors think that this judgement of the reviewer is strongly influenced by the misunderstanding regarding the purpose of this work, which had not been properly explained by the authors in the introduction of the original version of the paper. The introduction will be updated in order to impede such a misunderstanding for further readers (please see the new proposal for the introduction in the appendix of this letter of response).

*[6] Furthermore, there are no substantial new concepts, ideas, and methods brought forward by the authors. On the contrary, the description of BEM can be simply referenced by available textbooks (Manwell or Burton) as is the description of the extraction methods documented in Guntur and Sorensen.*

The reviewer is completely right concerning the section "2.1 Basic relations", which reproduces very basic relations from textbooks and which is included only for better readability. This section could be entirely removed and replaced by a reference to the book by M.O.L. Hansen ("Aerodynamics of wind turbines") or another textbook, maybe at the cost of readability, as the equations in this section are referenced many times in the subsequent sections of the paper. Section 2.2 on the annular sections method is very short and should, according to the authors' opinion, stay in the current paper for the sake of completeness.
Section 2.3 on inverse BEM includes important remarks concerning turbulent wake state correction, which cannot be found in the work by Guntur and Sorensen, and which should stay within the current paper in the authors' opinion.

*[7] There are no overall hypotheses, questions asked, or objectives for this work. The reviewer cannot apprehend the actual purpose/objective of this work and how the community can use the results obtained.*

This comment is, again, the result of a misunderstanding of the objectives of this work, which is most likely caused by the fact that the introduction of the original paper was not sufficiently clear on this issue. This will be improved in the revised version (see new introduction in the appendix).

The main objective of the proposed work is to perform a model update for BEM by using data from RANS simulations of a 3D rotor. This is successful if the results of the updated BEM (BEM-3D) are closer to the RANS results than the results of BEM with common 2D airfoil data (BEM-2D).

*[8] The scientific quality is also only FAIR because of several statements on rotor induced flow that are erroneous, see specific comments below.*

See the responses to the specific comments below.

*[9] The presentation quality is GOOD overall with minor suggestions to rewording.*
*In summary, the reviewer thinks that this work is of conference proceedings quality but does not merit publication in WES as a selected work from the TORQUE 2016 conference.*
*The reviewer hopes that the authors understand the many technical comments as constructive criticism, and the reviewer would hope to see further work by the authors at future technical meetings.*

The authors assure that they understand the reviewer's detailed comments as constructive criticism, and they are attempting to improve the work by accounting for his remarks as good as possible. Again, the authors want to express their regret that some of the criticism is the result misunderstanding the purpose of the investigated method, which can clearly be attributed to missing information in the original version of the paper, in particular in the introduction.

*[10] Specific comments and technical corrections:*
*Page 1, Line 8: Double "the the : : :"*
*Page 1, Line 15: A better word choice may be "attempted" (rather than "tried")*
*Page 1, Line 23: "models" and not "model"*

The proposed corrections are included in the new introduction (see appendix).

*[11] Page 2, Figure 1: Not sure if this figure is needed as it does not provide any new (or more clear) information as presented in the narrative.*

The figure will be removed.

*[12] Pages 3-4: These are very basic BEM relations that can be simply referenced from Hansen, Burton, or Manwell. No need to go through this.*
*Page 4, Line 17: Is 2014 the correct publication year?*
*Pages 5-6: Again, section 2.3 is documented in earlier work [Guntur and Sorensen]. The reviewer does not see the purpose of presenting the analysis/methodology if the reader can simply look at the original publication.*

The conference was in 2012, but the Conference Proceedings in the *Journal of Physics: Conference Series* have been published in 2014 according to the Journal's website. Other

sources, for example DTU's database ([http://orbit.dtu.dk](http://orbit.dtu.dk)), also list the publication under the year 2014. Therefore, the authors suppose that 2014 is the correct year. Regarding the reproduction of some content from the work by Guntur and Sorensen, please see the response to [6].

*[13] Page 6, Line 12: "… but it is used the other way round" is non-scientific language.*

The phrase will be replaced, for example "… but is used reversely …"

*[14] Page 7, Figure 3: It is difficult to discern from the contours but it seems that the axial induction factor a is increasing toward the tip, at least in parts (a) and (c). This is consistent with classical BEM and vortex methods and rejects some of the strong statements made by the authors, see below.*

Please see the response to [25] below.

*[15] Page 7, section 3.3: Confusing terminology of 'Glauert correction'. The authors refer to it as the high CT correction used in BEM methods, while other literature refers to the classical tip correction. The authors may reconsider to be more clear on this in the future.*

As the reviewer states correctly, when the authors say "Glauert correction", they refer to the correction which is applied in the case of high induction / high CT. This correction is referred to as "Glauert correction" in the book by M.O.L. Hansen [Ref 2], which is one of the main references of the paper. The authors admit that there a risk of misinterpretation as the tip correction by Prandtl and Glauert (which is also mentioned in several places in the paper). Therefore, the term "Glauert correction" in the sense of "high CT correction" will be replaced by "turbulent wake state correction", which is a terminus from e.g. Leishman's book [Ref 3] and which had already been given as an alternative name in the original paper (p. 6, l. 7).

*[16] Page 7, section 3.3: Reviewer statement: "If no tip correction is used in the inverse BEM analysis, then the 3D polar data MUST depend on the blade pitch angle". In the reviewer's opinion, this is a trivial result and one that is really not useful for BEM based efficient design tools. The reason why tip corrections exist is because they can account (at least to some extent for lightly loaded rotors) for the effects of solidity (blade number) and blade pitch angle (through the blade flow angle). The reviewer does not see any scientific merit here.*

As already mentioned above, in the proposed work it is attempted to replace as much modelling as possible by knowledge which is available from CFD. One of the reasons for deviations between RANS and BEM with tip correction is that the tip correction models are built on a number of assumptions which are not met in RANS or under realistic conditions. The authors want to present a result which resembles the CFD result as close as possible. For this reason, they wanted to use as few assumptions as possible in the BEM, and as much information as possible from CFD.

For the 3D rotor polars obtained by the annular sections method, the authors have compared the BEM results with and without Prandtl's tip loss factor (not shown in the paper; the figures for load case E2 are in Appendix 1 at the end of this letter of response). The result was that Prandtl's tip loss factor did only change the result slightly in the most outer part of the blade, whereas the deviations between BEM and CFD are significantly larger, even in the central part of the blade. The authors do not have any hint that this kind of tip loss correction can bring the

BEM and CFD results significantly closer together.

*[17] Pages 9-10, section 3.5: The higher the tip pitch angle, the lower the rotor thrust coefficient (which are not documented). It would have been useful to include integrated rotor thrust and power coefficients and to quantify differences beyond qualitative comparisons of sectional blade loads.*

Integrated rotor loads like thrust and torque (or power) are a direct consequence of the local rotor loads. The rotor thrust is simply the sum of the axial forces in the blade elements. If the local rotor loads are very close to each other in two methods, then the integrated loads will also be very close to each other. Similar integrated loads, however, are not a sufficient condition for similar local loads (there could be compensation of errors in different parts of the blade).
A comparison of the global thrust and torque on the rotor between the different methods for load case E2 is shown in Appendix 2 at the end of this letter of response.
This figure can be included in the revised version of the paper as an example for the integrated loads.

*[18] Pages 9-10, section 3.5: The authors do not provide a physical reasoning as to the behavior observed in Figure 5 rather than a plain description of what can be seen in the corresponding figure. The reviewer feels that this is a missed opportunity of providing some further physical insight into the aerodynamic behavior of the given turbine at that particular operating condition. Again, what is the purpose of this work?*

The purpose of this work is updating the BEM with data from CFD simulations of a rotor (as explained in more detail in the answer to comment [1]). A detailed analysis of the reasons for the pitch angle dependency of the 3D rotor polars would indeed be very interesting.
With a different pitch angle, the induction at the blade is different in magnitude and distribution (induction is smaller if the blades are pitched towards feather). This changes, more or less, all of the 3D effects on the blades (local inflow angles, crossflows, rotational augmentation, tip / hub vortices, …). In the context of BEM, all these 3D rotor effects have to be lumped together in one local angle of attack. This is quite a simple reduction for a number of rather complicated effects. Of course, these are only vague hypotheses which should be justified in a closer analysis. The authors think that this could be an opportunity for another publication after thorough investigation of the effects, but in the current paper, the focus should stay on updating the BEM with data from CFD simulations.

*[19] Page 10, Lines 7-9: "It is noted that in the common BEM with 2D airfoil coefficients, the lift and drag polars are always assumed to be independent of the pitch angle. However, the findings of the present work indicate that this is wrong for 2D polars as well as for 3D rotor polars". This is correct but an artifact of the extraction method in the sense that the inverse BEM methodology does not include any tip correction, see also earlier comment. While this is a correct finding by the authors, it is not surprising and, more importantly, does not add much scientific merit to the community as it is very unrealistic to use pitch corrections for airfoil tables in a general predictive sense, particularly when the tip loading is such that the flow is attached (pitch angles of 0deg, 10deg, 20deg).*

Application in blade design, e.g. during the selection of airfoils or chord and twist distribution, is not the objective of this work. Application in a predictive sense is possible for fixed geometry, e.g. for load simulations in the certification process. In this case, the application of pitch-dependent airfoil coefficients seems to be realistic to the authors. Indeed, the pitch angle dependency is not desirable, and it is a practical problem during application that usual BEM

codes do not support pitch-angle-dependent airfoil coefficients and their interpolation, but this does not make the investigated method unfeasible.

Please see the response to [1] for a statement on the purpose of this work, and to [16] for the authors' view on tip correction.

*[20] Again, it is not clear what is the actual purpose of this work and what the results should be used for?*

The purpose of this work is updating BEM models with data from 3D rotor CFD simulations. The intended scope of application are simulations in a late design stage, e.g. for certification, where at least a preliminary geometry is fixed and a CFD-based description of the aerodynamic forces is desired. Please see the response to comment [1] for a more detailed explanation. The authors regret that they did not make this clear in the original version of the paper. Please see the proposal for a new introduction in the appendix to this letter of response.

*[21] Page 12, Discussion of Figure 7: It would have been really useful to document integrated rotor thrust and torque coefficients to better understand the differences between the various methods. Again, no physical explanation is given; the authors merely discuss what the reader can see in the figures anyhow. Are there any convergence issues for the inboard blade sections?*

Regarding the integrated loads, please see the response to [17].

Regarding the possibility of convergence issues at the inboard sections, it is noted that the BEM algorithm converged in the sense that the axial induction factor did not change more than 1e-10 from iteration to iteration, except for the most inner section, where it does not reach this threshold within 500 iterations. The values of the BEM results can be explained by the values of the airfoil polars used in these simulations.

The RANS simulations, however, did show difficulties in convergence close to the blade root, as already mentioned on p. 8, l.28 and on p. 18, l. 3 of the original paper. These convergence issues in the RANS simulations occur for several different grid resolutions. It is well known to the CFD community that steady-state RANS simulations are likely to have convergence issues for flow situations which are intrinsically unsteady, as it is the case at the most inboard part of the blade, where the flow is separated even at relatively low wind speeds due to the high angle of attack and due to the flow disturbance by the hub. Unsteady simulations show oscillations in the local forces which indicate periodical vortex shedding, however, these unsteady results cannot be used within the current methodology and their presentation is outside of the scope of this paper.

*[22] Comment on Figures: Vertical axes would be best presented in [kN]. The behavior at the blade root is very concerning; the reviewer suspects that BEM solutions might not be fully converged.*

The labels at the axes will be presented in kN, or maybe in kN/m (per blade length) in the revised version. Regarding the possibility of convergence issues, please see the response to [21].

*[23] Page 13: Why is load case E2 presented prior to E1?*

Because the load cases E1 to E3 are numbered in the order of increasing wind velocity, and E2 is the case with rated conditions, i.e. an important case with very realistic flow conditions, whereas E1 is a case with very high tip speed ratio, and therefore high induction, which is necessary to produce high local angles of attack for the extraction of the 3D rotor polars, but is unlikely to be

found at a turbine during regular operation. The authors will consider switching the load cases E1 and E2 in order to avoid confusion.

*[24] Page 14, Figure 10: The behavior at the blade root is very concerning with respect to numerical accuracy and solution convergence of computed results in general.*

The BEM solutions did converge in the sense that the induction factor did not change more than 1e-10 from iteration to iteration, except for the most inner section, where it still changes in the order of 1e-9 even after 500 iterations. The values at the blade root are caused by the slope of the polars (in particular the 2D polars from DTU). Note that it is a difference of forces that is shown in these curves, so the slopes (not the values) of the polars determine the shape of the curve. Obviously, the slopes differ quite severely between the different sections. Also note that the behavior with the 3D rotor polars is significantly better than with the 2D polars.

*[25] Page 14: … reduction at the tip by increasing the induction factor, which is physically wrong." and "…, which is physically wrong and can be considered an inconsistency in the tip loss model by Prandtl/Glauert." Unfortunately, these strong statements are both very wrong and concerning as to the authors' understanding of basic blade aerodynamics.*
*1) The increase in the axial induction factor a toward the blade tip is very consistent with vortex methods (from which the Prandtl/Glauert tip correction is derived) and finite-wing lifting-line theory in general; the reviewer also wants to mention section 3.4.3 in Burton (Relationship between blade circulation and induced velocity) which can serve as a simple counter example to the authors' statements.*

The reviewer is right that increasing induction close to the tip is typical for BEM and vortex methods, however, this does not reject the statements made afterwards, as the BEM-3D curves in this figure were not created with conventional BEM (2D airfoil coefficient, tip loss model, …), but with 3D rotor polars from CFD. The results from BEM-3D match well with the CFD results. The authors know several CFD simulations from several different institutions which show a decrease of the induction factor close to the tip. One example is [Ref 1] (see end of this document). Figures 2 and 3 in [Ref 1] show the induction factor vs. radius which were evaluated from 3D rotor CFD simulations by various methods, and also induction factors from a simple BEM (ECN-BEM) and a vortex wake method (ECN-AWSM). Note that the figures on the right hand side refer to the INNWIND rotor, which is the same rotor as used by the authors in the current paper. It is clearly visible that the BEM and the vortex wake method produce increasing induction factors when approaching the tip, as expected by the reviewer based on his experience (so we guess) with vortex methods and based on the section in Burton's book. However, all CFD-based induction factor distributions which are known to the authors clearly show decreasing induction close to the tip, just as the result in figure 10 of the proposed paper does. To the authors' knowledge, increasing induction when approaching the tip is typical for vortex methods as well as for classical BEM, but not for CFD-based methods and therefore neither for BEM with 3D rotor polars.

The physical reason for decreased induction close to the tip might be that the induction factor is (almost) zero at a short distance outside the rotor radius, and as the induction factor (or the axial velocity) cannot jump from a large value to zero at once, it decreases when approaching the tip. The publication by Shen et al. (2005), which is cited in the original version of the paper (p. 14, l. 1), shows analytically that the axial induction factor approaches unity in BEM (for non-zero drag), which they call "unphysical". This motivated the strong statement that the reviewer criticizes. The authors admit that the term "unphysical" is inadequate in the context of the current paper because there is not any comparison to actual physics, only to RANS results. The term "unphysical" will therefore be replaced by "in contradiction to the RANS results".

*[26] 2) The reviewer does not understand why the 'a' distribution(s) from Figs. 3 and 6 cannot be used for comparison; in fact, they actually indicate an increase in 'a' toward the blade tip (maybe with the exception of Fig. 3b). This really raises questions as to the scientific quality of this work.*

The reason why the *a* distribution from RANS is not shown in these figure is, as stated in the original version of the paper, that "there is no unique way to determine the induction factor from RANS; it can only be obtained by e.g. the annular sections method or the inverse BEM method, but these results differ" (p. 13, l. 2f). With this remark, the authors want to express that the induction factor as required for comparison with BEM simulations is not a direct result of a RANS simulation. From a RANS simulation, the axial velocity component is obtained locally on every grid point in the volume. In order to obtain values for the induction factor which can be compared to BEM, there are various possibilities. Two methods are presented in the current paper: The annular sections method and the inverse BEM method.

A modified version of figure 10 including both variants for evaluation of the axial induction factor is in the appendix of this letter of response. This figure shows that the induction factor obtained from RANS by inverse BEM and the induction factor from *BEM-3D invBEM* match almost exactly. This is nothing more than a simple verification of the implementation of the method, and does not bring any benefit. Including the induction factor obtained from RANS with the annular sections method yields a curve that shows an offset from the *BEM-3D anSecs* curve. This offset is caused by the inconsistency between the annular sections method and the BEM algorithm in the calculation of the induction factor, which is explained on p. 12, l.16f in the original paper.

Both methods for evaluation of the induction factor (annular sections method and inverse BEM method) show a decrease of the axial induction factor when approaching the tip.

The authors will add some more explanatory sentences on this issue in the revised version.

The authors have double-checked the contour plots as shown in figs. 3 and 6 and the line plot in fig. 10 and could not find any error or inconsistency between these figures. Figure 10 shows the same case as fig. 3(b). The authors are not capable of judging from fig. 3(b) how fig. 10 should exactly look like; however, they cannot find any clues that would promote doubt in the consistency of the figures.

Including the modified figure as shown in appendix 3 to this letter of response might be a bit confusing because there are two RANS lines, and we do not know which of them is "the better one". Actually, none of them can be claimed to be correct or better than the other. The two lines are just the results of two different ways of evaluating the same 3D RANS result. One of it (the invBEM) is obviously consistent with BEM, whereas the other one (anSecs) is not. Leaving out the RANS curves and explaining the reason for this in the text, as done in the original version, was meant to facilitate understanding for the reader. As this obviously can lead to wrong conclusions and suspicions about the reasons for leaving them out, the authors would tend to include the figure as shown in the appendix of this letter of response into the revised version of the paper, or maybe split up this figure into two separate figures for more clarity.

*[27] Page 18, Figure 14: What is going on at the blade root? Are the solutions converged and to what accuracy ? This is very concerning.*

Please see the response to [24].

On behalf of the authors,
Marc S. Schneider

[Ref 1] H. Rahimi, M. Hartvelt, J. Peinke, J. G. Schepers: Investigation of the current yaw engineering models for simulation of wind turbines in BEM and comparison with CFD and experiment, J. Phys: Conf. Ser. 753, 2016, doi 10.1088/1742-6596/753/2/022016

[Ref 2] Hansen, M. O. L.: Aerodynamics of Wind Turbines, Earthscan London, second edition, 2008

[Ref 3]  Leishman, J. G.: Principles of Helicopter Aerodynamics, Cambridge University Press, 2nd edition, 2006

**Appendix 1: The influence of Prandtl's tip loss model on the BEM-2D and BEM-3D results**

[Figure]

[Figure]

**Appendix 2: Comparison of integrated thrust and torque**

[Figure]

[Figure]

**Appendix 3: Figure 10 with additional curves from RANS (cf. response to [26] for explanation)**

[Figure]

**Appendix 4: Proposal for new introduction (section 1) of the proposed paper, taking into account both reviewers' feedback**

**1 Introduction**

The Blade Element Momentum method (BEM) is a widely used method for wind turbine design and load simulations. The BEM method intrinsically contains the assumption of radial independence of the blade elements, i.e. the blade elements do not influence each other, and 3D phenomena like the downwash from tip or hub vortices can only be captured by additional, mostly empirical submodels. BEM usually uses two-dimensional airfoil data, which were obtained for cross sections of the blade by wind tunnel experiments or numerical simulations. These 2D airfoil data, together with some empirical models, are assumed to completely describe the aerodynamic behaviour of each blade element.

In order to account for 3D influence in the BEM, various correction methods have been proposed: For a better description of the rotational augmentation of aerodynamic forces and delayed stall close the blade root, models for correction of 2D airfoil coefficients were developed e.g. by Du and Selig (1997), Chaviaropoulos and Hansen (2000) and Dowler and Schmitz (2014). For the decrease of aerodynamic forces close to the blade tip (tip loss), the traditional model by Prandtl/Glauert (described e.g. in Hansen (2008)) is still widely used, although improved and new tip loss models have been proposed (e.g. Shen et al. (2005) or Sørensen et al. (2016)). Bak et al. (2006) present a more general approach which corrects 2D airfoil data based on semi-empirical relations for the pressure distribution.

On the other hand, there are methods which avoid additional submodels, and instead try to include all the aerodynamic behavior into airfoil coefficients. In some works in the context of the NREL phase VI rotor, e.g. Tangler (2002), airfoil coefficients are evaluated based on data from rotor experiments as well as on 2D airfoil data.

As high-fidelity CFD simulations of whole rotors are not too difficult to obtain nowadays, it is quite an obvious idea to use data from 3D rotor CFD simulations in order to improve the results of a BEM analysis. The most straightforward way to achieve this is to calculate airfoil coefficients from CFD simulations, which can then be applied in a BEM code. This way of transfer from an expensive high-fidelity method to a fast engineering model, which we refer to as *model updating*, should enable the BEM to produce results which are very similar to CFD results for a range of parameters.

The objective of this work is to perform a model update for BEM by using airfoil coefficients from 3D rotor RANS simulations (these coefficients will be called *3D rotor polars*). This means that BEM is treated as a kind of reduced-order modelling, in the sense that it is attempted to reproduce results from RANS simulations by a drastically simpler and faster model. While validation of CFD results is of outmost importance if the results are meant to be used for productive purpose, a validated CFD solution is not necessarily required for investigation of the feasibility and the limitations of this model update. Therefore, the validation of the RANS solution is not addressed in this work. Instead, the results of the BEM with 3D rotor polars are validated against a number of RANS simulations in the sense of a code-to-code validation. The intended application of the BEM with 3D rotor polars are simulations in a later design stage, for instance in the certification process, when the geometry of the rotor is already fixed. BEM with 3D rotor polars is not suitable for blade design, as the airfoil coefficients extracted from 3D rotor

simulations are specific to the blade geometry, and the feasibility of 3D rotor polars for different geometries than the one used in the RANS simulations is questionable and not investigated in the current paper.

Within this work, two methods for determination of a set of 3D rotor airfoil coefficients from a number of steady-state three-dimensional RANS simulations of a rotor are investigated and compared. This requires that the local angle of attack and the lift and drag coefficients are extracted from RANS solutions. The extraction of lift and drag coefficients and the corresponding angle of attack from CFD simulations by different methods has already been described e.g. by Bak et al. (1999), by Johansen and Sørensen (2004) or Guntur and Sørensen (2014). In the present work, two methods for the evaluation of the induction factors and the local angles of attack at the blade were tested and compared. The airfoil coefficients obtained by this procedure will be called *3D rotor polars,* and the BEM with 3D rotor polars will be referred to as *BEM-3D*, as opposed to BEM with common 2D airfoil coefficients, which is referred to as *BEM-2D*.

Most of the content of this paper has already been presented at "The Science of Making Torque from Wind" (TORQUE) 2016 in Munich (Schneider et al., 2016). In contrast to the original paper, the present work uses the more renowned reference wind turbine from the European project INNWIND.EU, developed at DTU, with a rotor diameter of about 178 meters and a rated power of 10 MW (cf. Bak et al. (2013)). In addition, section 4 contains some new material in which the difference between steady states is evaluated in order to assess the accuracy of the slope of the 3D rotor polars and their potential for application in unsteady BEM simulations.

New references (the others can be found in the original version of the paper):

J.L. Tangler: *The Nebulous Art of Using Wind-Tunnel Airfoil Data for Predicting Rotor Performance.* National Renewable Energy Laboratory, Golden, Colorado; AIAA 2002-0040

C. Bak,  P. Fuglsang, N.N. Sørensen, H. Aagaard Madsen,  W.Z. Shen,  J.N. Sørensen (1999): *Airfoil characteristics for wind turbines*. Denmark. Forskningscenter Risoe. Risoe-R; Nr. 1065(EN)

---

## Author Comment (AC2) · 31 Mar 2017

**Reply to the comment of anonymous referee number 2**

The authors want to thank the referee very much for his detailed comment.

The authors feel that there are major misunderstandings, probably caused by the introduction of the original paper, which was not clear enough or even misguiding (a draft for a new introduction is appended to this letter of response).

Many of the reviewer's comments aim on the reliability of the CFD solution. The authors have attempted to make clear in the introduction of the paper (and were obviously not very successful with this attempt) that they consider their work as a contribution to model updating, or reduced-order modelling, in the sense that they try to update a fast engineering model (BEM) with data from an expensive high-fidelity method (RANS, in this case). The data transferred from RANS to BEM are the 3D rotor polars. Two different (existing) methods for the extraction of 3D rotor polars from CFD solutions are investigated. Subsequently, the 3D rotor polars are applied in BEM simulations. The procedure is successful if the BEM with 3D rotor polars produces similar results as the RANS simulations (this is verification for the operating points at which the polars were extracted, and code-to-code validation for different operating points). In this context, it is not necessary that the CFD solution is very accurate (in the sense of "close to reality"). We assume (without proving it) that our RANS solution is more reliable (closer to "the truth") than the BEM with common 2D airfoil coefficients and empirical corrections. When applying the method, it is of course desirable to use a thoroughly validated RANS setup and solution (although obviously no CFD simulation will ever be "the truth"), but the validation of the CFD simulation is a complex task for itself and therefore not in the scope of the proposed paper. The authors will make this clearer by extending and rewriting the corresponding paragraph (p. 2, l. 11ff) in the revised version of the paper (see the appendix of this letter of response).

Furthermore, it is very important to note that the methodology of BEM with 3D rotor polars is not suitable for blade design, because the 3D rotor polars depend on (and are possibly very specific to) the geometry of the rotor blade. This has always been conscious to the authors, but it has not been properly documented in the proposed paper. The applications which the authors had in mind are simulations in a later design stage, where a (preliminary) geometry is already fixed and a large number of load simulations is required, for example in the certification process. Only in this stage, it is worth the effort to create a RANS setup, perform RANS simulations and obtain a CFD-based, but very fast aerodynamic model (BEM with 3D rotor polars) which can be applied in a large number of simulations. This important explanation is missing in the original version of the paper. The introduction will be rewritten in the revised version of the paper in order to clarify this (see appendix).

In the following, the authors have split the reviewer's comment in several items and responded to every single item. The original reviewers comment is reproduced in *blue italic* letters.

Numbering in square brackets at the beginning of each paragraph has been added by the authors.

*[1] This paper is very relevant because it discusses the validity of the widely used BEM model and its input. Polars have been extracted from 3D CFD computations and compared to 2D polars*

*from e.g. 2D CFD computations. However, some important information is missing. We need to know whether the solver includes free transition and how big the CFD domain is.*

In the RANS simulation, the flow is fully turbulent, so there is no transition modelling included. The CFD domain (mesh) has the shape of a cylinder with a radius of 6 $R$ and an axial extension from 4 $R$ upstream to 10 $R$ downstream of the rotor, where $R$ is the radius of the rotor. These pieces of information will be added to section "3.2 RANS setup" of the paper.

*[2] Furthermore, it is assumed that the 3D CFD computations are correct and represents the "truth". Couldn't this be questioned? My experience is that (3D) CFD does not capture stall correctly. So do we believe in the stations where we have separated flow?*

As noted in the introduction to the original paper (p.2, l. 11ff), and as explained in more detail at the beginning of this letter of response, our objective is to do a kind of model updating / reduced order modelling. The intention is to use 3D rotor polars in BEM in order to obtain within seconds a result which is similar to a result from CFD which would require many hours or days to be obtained. A realistic (and validated) CFD solution is very desirable and definitely essential for practical application of this method, but it is not required for the investigation of the methodology which is done in the proposed paper.

*[3] And forces can be too high if the domain is too small and the flow has been pushed through the rotor because of the boundaries of the domain. The domain boundaries probably have to be 10 to 20 rotor diameters away.*

Please see the response to [1] for the distance of the boundaries of the CFD domain. Simulations with a larger domain did not show significant differences. The forces from our RANS simulations are in the correct order of magnitude (please also see the response to [5]). Anyway, as explained above, an accurate RANS solution (in the sense of "close to reality") is not required for the investigation within the proposed paper.

*[4] Furthermore, the extraction of the data can also be questioned. Was the radial component extracted?*

The radial velocity can be extracted, but it is not used in the proposed paper. Please see the response on [18] for a more detailed statement.

*[5] Also, there is something with the values where force distributions are shown. They are about 10 times too high if the forces are Newton per meter.*

As the axis labels indicate, the forces are in Newton, not in Newton per meter. This is suitable in this case as we use blade elements of equal length. The length of the blade elements is 2.87833 meters (rotor radius 89.15 m minus hub radius 2.8 m, divided by 30 (number of blade elements)). We consider changing the plots to N/m in order to avoid confusion.

*[6] Finally, the conclusions are not really conclusions but self-fulfilling statements since it is obvious that data based on 3D CFD compares better to 3D CFD.*

Please see the responses to [19] and [20] for statements on the conclusions.

*[7] Since the overall aim of our research is to provide data and methods for the further development of wind turbines, we have to wear the perspective of the wind turbine manufacturers. Would I as a manufacturer blindly believe the 3D CFD computations? I think not.*

Neither would the authors blindly believe in CFD computations (and not in BEM, vortex wake or any other type of simulation either). Validation of CFD simulations (by experimental data) is a task of outstanding importance. It requires the availability of suitable experimental data, and it is quite a large effort. This task is too large to fit into the projects on which the authors are currently working. In the context of the proposed paper, it is not necessary to show the validity of the CFD simulations. All methods in the paper can be applied to any CFD simulation, no matter whether they are believed to be valid or not. The intention of the proposed paper is (only) to investigate methods for the transfer of information from high-fidelity simulations (RANS) to fast engineering models (BEM) in the sense of model updating.

*[8] Therefore, I do not think that the advice should be to extract polars directly from 3D CFD. However, I think it could be more interesting to discuss why we see these differences between 2D and 3D.*

The authors completely agree that an in-depth analysis of the causes for the deviations would be very interesting. For the proposed paper, however, the focus is on model updating / reduced-order modelling, i.e. enhancing (or updating) a fast engineering method (BEM) by data from an expensive high-fidelity simulations (RANS). For this reason, in the paper we only say very roughly that the differences are due to 3D effects (tip or hub vortex, centrifugal forces, downwash, 3D induction distribution, etc.). A detailed investigation of the causes for the difference between BEM-2D and BEM-3D (or RANS) is not the purpose of the paper. For such an analysis, it would be better to do a thorough validation of the CFD result first (in particular at the critical positions, such as the inboard region where the flow is partly separated and heavily three-dimensional). However, this is outside of the scope of the proposed paper.

As there are several published methods for the extraction of the angle of attack from CFD results, their application for creating 3D rotor polars and using them in a BEM analysis is quite a logical consequence. To the authors' knowledge, there is hardly any literature where not only the extraction of 3D rotor polars, but also their application is investigated. With the proposed paper, the authors might contribute to close this gap in current literature. The analysis in the proposed paper shows that the application of 3D rotor polars can in principle work very well, but that it also brings several difficulties, as for example the pitch angle dependency of the polars. The authors are not conscious of a publication where this has been described before, and they believe that such findings deserve publication to the wind energy community.

*[9] Why do we see this AOA shift for different pitch angles? I am left with the feeling that you (and we) are overlooking something. How do we interpret the abstraction of AOA in 3D flow?*

We are not sure how you mean "interpret the abstraction of AOA in 3D flow", but in the context of this work, the AoA is calculated using the induction factors in the manner of the BEM (equations (5) and (6) in the paper), and the induction factors are obtained from the annular sections method or the inverse BEM method. This could be considered a local 2D-equivalent AoA, based on the induction factors from 3D CFD, but dependent on the method for extraction of the induction factors. We do not have a simple explanation for the pitch angle dependent shift in the polars, except for the vague statement that it has to do with different induction (magnitude and distribution) for different pitch angles, and hence different 3D influence (e.g. crossflows and rotational augmentation close to the hub, where we have the largest shift). See also the response to [21].

*[10] I propose the following should be considered in the paper:*
*\* look into the values of the forces, where I think there is an error*
*\* describe the CFD setup in more details (domain size, free transition etc) and*
*\* analyse where we see differences to 2D data and why you observe the shifts in angle-of-attack*
*\* change the conclusions so that it is not self-fulfilling statements.*

The values of the forces are correct, in particular their order of magnitude, cf. response to [5]. Some words on the CFD setup will be added (cf. response to [1]). We will see if we can find some more explanations for the differences between 3D and 2D airfoil data as well as the pitch angle dependency (although the investigation of the reasons is not the main aspect of this paper) and we will consider revising the conclusion (cf. response to [19] and [20]) in order to avoid what you call self-fulfilling statements.

*[11] I do not think that you should propose to use polars extracted directly from 3D CFD, because you then make the assumption that 3D CFD is correct.*

As for any simulation, the results obtained with this method need to be critically reviewed and – if possible – validated by comparison with appropriate experimental data. If you use common BEM with 2D airfoil coefficients, then you are assuming that BEM with 2D airfoil coefficients is correct. Validation is always required, whether using 3D rotor polars or not, but it is not the subject of the proposed paper. In the proposed paper, the focus is on model updating, i.e. bringing the BEM results close to the RANS results by using information from RANS. Please also see the responses to [2] and [7].

*[12] Specific comments*
*Abstract Line 8: "the the"*

This will be corrected, thanks.

*[13] Chapter "Introduction" \* I think you are missing one of the first attempts to make such airfoil characteristics from CFD: – Bak, C., Fuglsang, P., Sørensen, N. N., Aagaard Madsen,*

H., Shen, W. Z., & Sørensen, J. N. (1999). *Airfoil characteristics for wind turbines. (Denmark. Forskningscenter Risoe. Risoe-R; No. 1065(EN))*.

Thanks for the remark; a reference was added to the introduction for the sake of completeness (see new introduction in the appendix).

*[14] Section "Inverse BEM" Last sentence: I do not really understand*

Assuming that you mean the sentence "There is no need to do a post-processing of the volume solution of the RANS, as there is in the annular sections method for the calculation of the averaged axial and tangential velocities." (p. 6 l. 19f): This is a rather technical remark. There are CFD solvers (like TAU) which can output the solution on the surfaces of the domain (surface solution) separate from the solution on the whole grid (volume solution). The surface solution does not contain many points and is therefore quite a small file which is easy and fast to handle. The volume solution, in contrast, can have a file size of many Gigabytes and it may take some time and need some resources to copy, read and post-process such files. The inverse BEM requires as input only the forces on the surface, which are contained in the surface solution file, whereas the annular sections method requires a post-processing of the volume solution file, which is a bit less convenient. We will clarify or delete this sentence in the revised version in order to avoid confusion.

*[15] Section "RANS setup" How big is the domain? How far upstream? Downstream? And in radial direction? If the domain is small this can influence the result*

Please see the response to [1]. The proposed paper does not have the intention to discuss the validity of the CFD results, cf. the detailed explanation at the beginning of this letter of response.

*[16] Section "Influence of pitch angle" Figure 5: Title of plot is not clear.*

We forgot to update the title of the plots after removing the 20 deg polars (for the sake of clarity; with the 20 deg polars there are too many symbols and lines). We will correct the title, e.g. "Comparison of the Cl values obtained at pitch 0° and 10°".

*[17] Section "Comparison between Rans, …" Fig 7: What does the forces represent? N/m? If so: An integration of the tangential forces result in a power delivered by the blade of around 30MW – and for 3 blade 90MW. Can this be right? A factor of 10? And the same is the case for the axial loading*

The forces are in Newton (not normalized, cf. response to [5]) and for the whole rotor (i.e. multiplication by the number of blades is already done). The sum of the local axial forces in figure 7a (sum of the values at the dots) is around 1.82 MN (1.82e6 N), which appears to be a reasonable order of magnitude. This value matches with the values reported by different parties in a technical report / deliverable from the EU project AVATAR (cf. [Ref. 1] at the end of this letter of response, figures 22 and 23; note that in this report, according to the authors' interpretation, a case with wind 11 m/s and rotation speed 8.836 RPM was simulated (cf. Table

2 in the AVATAR report) instead of 9.6 RPM in our paper, so the resulting global force in our figure 7a is larger than in the AVATAR report, but it is definitely in the same order of magnitude).

The sum of the local torque contributions (values of the dots in figure 7b times local radius and sum over all these values) is around 9.51 MNm. This value multiplied by the rotational frequency Omega = 1.00531 1/s leads to an aerodynamic power of around 9.6 MW, which seems to be realistic to the authors as the rated power of the turbine is 10 MW and the wind at this operating point (11 m/s) is still a bit below the rated wind speed (11.4 m/s).

*[18] You extract axial and tangential induction. What about radial? Couldn't this explain some of the AOA shifts?? Or what explains the AOA shift for different pitch angles?*

We are able to extract the local radial velocities, but we did not use them in the analysis. We are not sure how the reviewer would define a radial induction factor (specifically, which reference velocity should be used for its definition). Common BEM theory does not account for radial induction, and as we want to do a model update (i.e. tuning the fast model (BEM) with high-fidelity (RANS) data such that the result of the BEM gets closer to the RANS result), instead of inventing a new variant of BEM, we leave the BEM algorithm itself as it is, i.e. without accounting for radial induction. However, the comparison of the radial velocities between the different load cases for explanation of the pitch angle dependence seems to be interesting. We will make and attempt in this direction and maybe add some remarks on that in the revised version of the paper.

*[19] Section "Conclusion"*
*\* First finding:  – It is obvious that polar data obtained from 3D CFD agrees better to 3D CFD than data not obtained from 3D CFD. So that is not really a conclusion. It have to be so – otherwise you have been inconsistent.*

The intention of this bullet point was to state that the polar extraction and BEM with 3D rotor polars has been applied successfully. This is pure verification of the implementation for the load cases which had been used for the extraction of the polars (of the cases shown in the paper, these are E1 to E3); however, this is more than pure verification for the load cases which had not been used for polar extraction (P1 to P3 in the paper). For cases P1 to P3, the comparison should be considered a code-to-code validation. The reviewer is right that this is not really a conclusion, but rather a part of the result. This first item should therefore not appear as a bullet point, but rather be included in the very brief summary paragraph at the beginning of section 5 "Summary and conclusions".

*[20] \* Second finding: – This is also obvious because you use the inverse BEM. So this is not either a conclusion*

The authors think that this is actually worth mentioning. To our knowledge, there is so far no literature in which the application of 3D rotor polars obtained by different methods is investigated. For instance, in the work by Guntur and Sorensen (2014) (from TORQUE 2012), the annular sections / AAT method was used for determination of the angle of attack, but there was

no subsequent application of the obtained 3D rotor polars in a BEM code, which is why the inconsistency of the annular sections method to the application in BEM was probably not found, or at least not reported in the publication by these authors. Therefore, we consider this to be a new finding and a contribution of the paper. In the beginning, we used only the annular sections method for obtaining the 3D rotor polars, and we were quite surprised by the offset in the forces which we saw in the results. The interesting aspect is rather not that the inverse BEM is consistent to the BEM, but that the annular sections / AAT method is not. It is possible that the inconsistency of the annular sections method to the BEM algorithm is obvious for some people, but we doubt that everyone will see this inconsistency at the very first glance, and it is (to our knowledge) not documented in literature so far.

*[21] * Third finding: – I would actually like to know WHY the polars change with pitch angle. Please consider a bit more.*

This is indeed a very interesting question, on which the authors do not have a simple answer. With a different pitch angle, we have different induction (less induction if the blades are pitched towards feather). This changes, more or less, all of the 3D effects on the blades (induction, crossflows, rotational augmentation, tip / hub vortices, …). In the context of BEM, all these 3D rotor effects have to be lumped together in one local angle of attack. This is quite a simple reduction for a number of rather complicated effects. Of course, these are only vague hypotheses which should be justified in a closer analysis. The authors think that this could be an opportunity for another publication after thorough investigation of the effects, but in the current paper, the focus should stay on updating the BEM with data from CFD simulations.

On behalf of the authors,

Marc S. Schneider

[Ref. 1] Niels N. Sørensen, Martin O.L. Hansen, Néstor Ramos García, Liesbeth Florentie, Koen Boorsma: *Power Curve Predictions WP2 Deliverable 2.3*, AVATAR, available online: http://www.eera-avatar.eu/fileadmin/mexnext/user/report-d2p3.pdf

**Appendix: Proposal for new introduction (section 1) of the proposed paper, taking into account both reviewers' feedback**

**1 Introduction**

The Blade Element Momentum method (BEM) is a widely used method for wind turbine design and load simulations. The BEM method intrinsically contains the assumption of radial independence of the blade elements, i.e. the blade elements do not influence each other, and 3D phenomena like the downwash from tip or hub vortices can only be captured by additional, mostly empirical submodels. BEM usually uses two-dimensional airfoil data, which were obtained for cross sections of the blade by wind tunnel experiments or numerical simulations. These 2D airfoil data, together with some empirical models, are assumed to completely describe the aerodynamic behaviour of each blade element.

In order to account for 3D influence in the BEM, various correction methods have been proposed: For a better description of the rotational augmentation of aerodynamic forces and delayed stall close the blade root, models for correction of 2D airfoil coefficients were developed e.g. by Du and Selig (1997), Chaviaropoulos and Hansen (2000) and Dowler and Schmitz (2014). For the decrease of aerodynamic forces close to the blade tip (tip loss), the traditional model by Prandtl/Glauert (described e.g. in Hansen (2008)) is still widely used, although improved and new tip loss models have been proposed (e.g. Shen et al. (2005) or Sørensen et al. (2016)). Bak et al. (2006) present a more general approach which corrects 2D airfoil data based on semi-empirical relations for the pressure distribution.

On the other hand, there are methods which avoid additional submodels, and instead try to include all the aerodynamic behavior into airfoil coefficients. In some works in the context of the NREL phase VI rotor, e.g. Tangler (2002), airfoil coefficients are evaluated based on data from rotor experiments as well as on 2D airfoil data.

As high-fidelity CFD simulations of whole rotors are not too difficult to obtain nowadays, it is quite an obvious idea to use data from 3D rotor CFD simulations in order to improve the results of a BEM analysis. The most straightforward way to achieve this is to calculate airfoil coefficients from CFD simulations, which can then be applied in a BEM code. This way of transfer from an expensive high-fidelity method to a fast engineering model, which we refer to as *model updating*, should enable the BEM to produce results which are very similar to CFD results for a range of parameters.

The objective of this work is to perform a model update for BEM by using airfoil coefficients from 3D rotor RANS simulations (these coefficients will be called *3D rotor polars*). This means that BEM is treated as a kind of reduced-order modelling, in the sense that it is attempted to reproduce results from RANS simulations by a drastically simpler and faster model. While validation of CFD results is of outmost importance if the results are meant to be used for productive purpose, a validated CFD solution is not necessarily required for investigation of the feasibility and the limitations of this model update. Therefore, the validation of the RANS solution is not addressed in this work. Instead, the results of the BEM with 3D rotor polars are validated against a number of RANS simulations in the sense of a code-to-code validation. The intended application of the BEM with 3D rotor polars are simulations in a later design stage, for instance in the certification process, when the geometry of the rotor is already fixed. BEM with 3D rotor polars is not suitable for blade design, as the airfoil coefficients extracted from 3D rotor

simulations are specific to the blade geometry, and the feasibility of 3D rotor polars for different geometries than the one used in the RANS simulations is questionable and not investigated in the current paper.

Within this work, two methods for determination of a set of 3D rotor airfoil coefficients from a number of steady-state three-dimensional RANS simulations of a rotor are investigated and compared. This requires that the local angle of attack and the lift and drag coefficients are extracted from RANS solutions. The extraction of lift and drag coefficients and the corresponding angle of attack from CFD simulations by different methods has already been described e.g. by Bak et al. (1999), by Johansen and Sørensen (2004) or Guntur and Sørensen (2014). In the present work, two methods for the evaluation of the induction factors and the local angles of attack at the blade were tested and compared. The airfoil coefficients obtained by this procedure will be called *3D rotor polars,* and the BEM with 3D rotor polars will be referred to as *BEM-3D*, as opposed to BEM with common 2D airfoil coefficients, which is referred to as *BEM-2D*.

Most of the content of this paper has already been presented at "The Science of Making Torque from Wind" (TORQUE) 2016 in Munich (Schneider et al., 2016). In contrast to the original paper, the present work uses the more renowned reference wind turbine from the European project INNWIND.EU, developed at DTU, with a rotor diameter of about 178 meters and a rated power of 10 MW (cf. Bak et al. (2013)). In addition, section 4 contains some new material in which the difference between steady states is evaluated in order to assess the accuracy of the slope of the 3D rotor polars and their potential for application in unsteady BEM simulations.

New references (the others can be found in the original version of the paper):

J.L. Tangler: *The Nebulous Art of Using Wind-Tunnel Airfoil Data for Predicting Rotor Performance.* National Renewable Energy Laboratory, Golden, Colorado; AIAA 2002-0040

C. Bak,  P. Fuglsang, N.N. Sørensen, H. Aagaard Madsen,  W.Z. Shen,  J.N. Sørensen (1999): *Airfoil characteristics for wind turbines*. Denmark. Forskningscenter Risoe. Risoe-R; Nr. 1065(EN)